# Efficacy of High-Ozonide Oil in Prevention of Cancer Relapses Mechanisms and Clinical Evidence

**DOI:** 10.3390/cancers14051174

**Published:** 2022-02-24

**Authors:** Alberto Izzotti, Enzo Fracchia, Camillo Rosano, Antonio Comite, Liliana Belgioia, Salvatore Sciacca, Zumama Khalid, Matteo Congiu, Cristina Colarossi, Giusi Blanco, Antonio Santoro, Massimo Chiara, Alessandra Pulliero

**Affiliations:** 1Department of Experimental Medicine, University of Genoa, 16132 Genoa, Italy; 2IRCCS Ospedale Policlinico San Martino, 16132 Genoa, Italy; camillo.rosano@hsanmartino.it (C.R.); liliana.belgioia@unige.it (L.B.); 3Galliera Hospital, 16128 Genoa, Italy; enzo.fracchia@galliera.it; 4Laboratory of Electron Microscopy, Department of Chemistry and Industrial Chemistry, University of Genoa, 16146 Genoa, Italy; antonio.comite@unige.it; 5Department of Health Sciences, University of Genoa, 16132 Genoa, Italy; zumama.khalid@edu.unige.it (Z.K.); 3370203@studenti.unige.it (M.C.); alessandra.pulliero@unige.it (A.P.); 6Mediterranean Institute of Oncology (IOM), 95029 Catania, Italy; salvatore.sciacca@grupposamed.com (S.S.); cristina.colarossi@grupposamed.com (C.C.); giusi.blanco@grupposamed.com (G.B.); 7UO Neurosurgery, Hospital Umberto I, 00161 Rome, Italy; antonio.santoro@uniroma1.it (A.S.); massimo.chiara@uniroma1.it (M.C.); 8Department of Surgery, La Sapienza University, 00185 Rome, Italy

**Keywords:** cancer prevention, cancer therapy, ozonized oil, prevention of cancer relapses, oxidation

## Abstract

**Simple Summary:**

Cancer relapses after chemo-radiotherapies arise from cancer stem cells able to escape cell killing because of their high antioxidants level. The aim of this study was to test the efficacy of ozonized oils to decrease the rate of cancer relapses. In vitro, oils at high ozonide content penetrate inside cancer cells releasing oxygen and reactive oxygen species damaging the thin outer membrane of inactive mitochondria. This event triggers intracellular calcium release and activates apoptosis. In vivo, ozonized oil has been administered by the oral route effectively decreasing blood antioxidants in cancer patients. This approach results in significant increase of survival rate and decrease of relapses in 115 cancer patients (brain, lung, pancreas, colon, skin) undergoing standard radio-chemotherapy regimens during a 4-years follow up. Obtained results indicate that the administration of ozonized oil represents an integrated approach to decrease the risk of radio-chemoresistance and cancer relapses in cancer patients.

**Abstract:**

Background: Cancer tissue is characterized by low oxygen availability triggering neo angiogenesis and metastatisation. Accordingly, oxidation is a possible strategy for counteracting cancer progression and relapses. Previous studies used ozone gas, administered by invasive methods, both in experimental animals and clinical studies, transiently decreasing cancer growth. This study evaluated the effect of ozonized oils (administered either topically or orally) on cancer, exploring triggered molecular mechanisms. Methods: In vitro, in lung and glioblastoma cancer cells, ozonized oils having a high ozonide content suppressed cancer cell viability by triggering mitochondrial damage, intracellular calcium release, and apoptosis. In vivo, a total of 115 cancer patients (age 58 ± 14 years; 44 males, 71 females) were treated with ozonized oil as complementary therapy in addition to standard chemo/radio therapeutic regimens for up to 4 years. Results: Cancer diagnoses were brain glioblastoma, pancreas adenocarcinoma, skin epithelioma, lung cancer (small and non-small cell lung cancer), colon adenocarcinoma, breast cancer, prostate adenocarcinoma. Survival rate was significantly improved in cancer patients receiving HOO as integrative therapy as compared with those receiving standard treatment only. Conclusions: These results indicate that ozonized oils at high ozonide may represent an innovation in complementary cancer therapy worthy of further clinical studies.

## 1. Introduction

Cancer cells differ from normal cells in several aspects, among which the blockage of the mitochondrial function stands out. Mitochondrion is the main endogenous source of oxidizing molecules. The mitochondrial blockage occurring in cancer cells is known as the Warburg effect [1]. Recent experimental research provided evidence that cancer cells are favored in their growth by antioxidant molecules but contrasted by pro-oxidant molecules. Cancer stem cells, which give rise to chemo/radio resistance and relapses, are characterized by a reducing environment, therefore being sensitive to the cytotoxic effects of oxidative damage [2]. Cancer tissue is characterized by a very low level of lipid peroxidation compared with the surrounding healthy tissues, as shown by biopsy analysis of 120 patients with primary hepatocarcinoma [3]. Accordingly, today the increase in oxidative damage represents a possible strategy for cancer therapy [4]. Intracellular generation of reactive oxygen species has been proposed as a possible Trojan horse for eliminating cancer cells [5].

Several attempts have been made in order to use ozone as a source of oxidizing species in cancer therapy. Ozone was used as a gas [6] or as an ozonized aqueous solution [7]. These studies reported specific cytotoxic effects of ozone on cancer tissues without any damage in healthy tissues. However, the therapeutic effect obtained was meaningful but transitory. When used as a gas or aqueous solution, ozone, due its volatility, displays transitory effects only in the extracellular environment. Cells are surrounded by a lipophilic membrane hampering the entry of gas or water. Cancer cells are well equipped with antioxidants molecules to counteract cancer chemo/radio therapies. Accordingly, an oxidative intracellular environment may be effective in counteracting cancer chemo/radioresistance. Furthermore, the setting up of a high level of oxygen in the cancer mass may be useful for preventing metastases development. Indeed, low oxygen availability (hypoxia) is the main mechanism triggering the migration of cancer cells from the primary site. Hypoxia promotes cancer invasion and metastatisation by activating the *met* oncogene [8]. Hyperbaric ozone has been proposed as a possible tool for preventing these events [9].

In this context, microRNA-based epigenetic regulation plays an important role. MicroRNA alterations drive cancer cells towards chemo/radioresistance [10] and modulate oxidative stress in cancer cells [11]. miR-146 plays a fundamental role in lung cancer progression and chemoresistance [12]. 

Increase in antioxidants such as reduced glutathione induces multi-drug resistance in neuroblastoma [13]. Downregulation of miR-15 and miR-16 is associated with increased availability of reduced glutathione in neuroblastoma cancer cells, contributing to chemoresistance [14].

The antioxidant environment characterizing cancer cells is related to the constitutive overexpression of Nrf2; the pivotal transcription factor regulating the activation of antioxidant response elements [15]. Nrf2 activity provides growth advantage by increasing cancer chemoresistance and enhancing tumor cell growth [16]. Overexpression of Nrf2 in cancer cells protects them from the cytotoxic effects of anticancer therapies, resulting in chemo- and/or radioresistance [17].

In this context, the herein presented study aimed at evaluating the efficacy of pharmacological preparations composed of ozonized oils in counteracting the growth of cancer cells. Ozonized oils are very stable molecules displaying antimicrobial properties [18]. The main goal of our study was the development of high-ozonide ozonized oil (HOO) to deliver a high amount of ozone-derived oxidizing species in a lipophilic complex able to penetrate the cancer cells and to activate apoptosis without damaging healthy tissues. Ozonized oils consist of unsaturated fatty acids that have been subjected to the action of ozone. The ozone is added to the double carbon–carbon bonds, with the formation of molozonides. These molecules quickly rearrange themselves according to the Criegee mechanism, causing the formation of trioxanes. Ozonides are generally unstable, while trioxanes are relatively stable but decompose under the action of reducing agents or intracellular enzymes. When the addition of ozone to the oil reaches the saturation of the double bonds, the viscosity increases with the progressive formation of ozonides until the oil reaches the consistency of gelatin. The peroxides contained in the oil can be hydrolyzed, giving rise to aldehydes and ketones with shorter chains compared with the original fatty acid. The length of the residues is determined by the position of the double bond along with the chain that reacted with ozone. 

The goals of the herein presented experimental study were the evaluation of HOO: (a) anticancer efficacy in vitro in cultured cancer cells; (b) molecular mechanism of action at the intracellular level; (c) mechanism of action at the systemic level; (d) inability to damage non-cancer cells; (e) anticancer efficacy and safety in vivo in human subjects and cancer patients.

## 2. Materials and Methods

### 2.1. Experimental Evidence in Cultured Cancer Cells

HOO was tested in human lung adenocarcinoma cells (A549 cell line) (IRCCS Policlinico San Martino, Genoa, Italy), and they were grown in D-MEM (GIBCO Invitrogen, Milano, Italy) containing 10% fetal bovine serum (Sigma-Aldrich, Milano, Italy) at 37 °C in 5% CO_2_ and 100% humidity. Non-ozonized sunflower oil was used as comparative sham control. The experiment was performed in quadruplicate. 

Glioblastoma U87MG cells were made available from the Biological Bank and Cell Factory of the IRCCS Ospedale Policlinico San Martino, Genoa, Italy. They were grown in DMEM high glucose media (Sigma-Aldrich, Milan, Italy), supplemented with 10% fetal calf serum (Euroclone, Milan, Italy), 2 mM L-glutamine (Euroclone, Milan, Italy), and 1% penicillin–streptomycin (Euroclone, Milan, Italy) at 37 °C in 5% CO_2_ incubator. 

Cells were seeded in 96-well plates at a density of 6 × 10^3^ cell per well in 100 uL of culture medium, and they were treated with 10% HOO for 2, 6, 12, and 24 h. Sham-treated cells were used as control. After treatment, cells were washed with PBS (Euroclone, Milan, Italy), fixed and stained with a solution of crystal violet containing methanol 20% *v*/*v*. The following day, a solution of acetic acid 30% *v*/*v* in water was added, and samples were read by a microplate photometer (Multiskan FC, Thermo Scientific) at 570 nm.

Cell viability was determined by Trypan blue staining (labeling dead cells) and MTT test (labeling viable cells), as previously reported [19]. 

### 2.2. Pharmacological Mechanism. Experimental Evidence in Lung Cancer Cells

The dynamics whereby the HOO formulation induces the killing of cancer cells were examined using a normal and trichrome fluorescence microscopy. The nucleus was stained in blue by DAPI (Sigma), the mitochondrial membranes in green by DiOC6 (Sigma), and calcium release into cytoplasm in red by Rhodamine2 (Sigma). HOO was stained by red Nile dye (Sigma) to trace its penetration inside cell cytoplasm.

### 2.3. Field Emission Scanning Electron Microscopy and X-ray Diffraction Analyses

The HOO mechanism of killing cancer cells was also explored by a field emission scanning electron microscope (FE-SEM, Zeiss Supra 40VP, Carl Zeiss, Germany) equipped with energy dispersive X-ray analysis (EDX) microprobe for elemental analysis (Oxford “INCA Energie 450 × 3”, Oxford Instruments, UK), comparatively examining sham-treated and HOO-treated A549 cancer cells. EDX elemental analysis was performed at high magnifications (10,000×) with a spot at the middle of the cells before and after the HOO treatment.

### 2.4. Evaluation of Apoptosis

HOO was added to cell culture medium in order to discriminate the prevalent cell death mechanism between necrosis and apoptosis. A549 human lung adenocarcinoma cells were purchased from the Biological Bank and Cell Factory (IRCCS Policlinico San Martino, Genoa, Italy). They were grown in DMEM medium (Sigma-Aldrich, Milan, Italy), supplemented with 10% fetal calf serum (Euroclone, Milan, Italy), 2 mM L-glutamine (Euroclone, Milan, Italy), and 1% penicillin–streptomycin (Euroclone, Milan, Italy) at 37 °C in a 5% CO_2_ incubator. 

The day before the experiment, A549 cells were seeded in a 6-well plate at a density of 8 × 10^4^ cells per well in 3 mL of culture medium DMEM (Sigma-Aldrich, Milan, Italy). After twenty-four hours of seeding, cells were treated with 10% *v*/*v* of HOO for 2 h and 4 h. Then, Muse™ Annexin V & Dead Cell Assay was performed. Cells were dissociated from each well to obtain single-cell suspensions, and 100 μL of these suspensions was added to each tube, together with 100 μL of the Muse™ Annexin V & Dead Cell Reagent (BD Biosciences Pharmingen 2350 Qume Drive San Jose, CA, USA). The samples were mixed thoroughly by vortexing and then stained at room temperature in dark for 20 min before being analyzed by flow cytometry (FACS Canto II cytometer, Becton Dickinson BD, Franklin Lakes, NJ, USA).

Microscope examination of cell morphology showed that in cancer cells treated with ozonized oil, at 1 h cell viability is still maintained, while cell sufferance and lack of viability is massive at 24 h. Accordingly, the mechanisms causing loss of cell viability should occur in the 1–24 h time interval. Because the activation of apoptotic mechanisms requires at least 4 h, this was the timeline when we decided to evaluate this parameter.

### 2.5. Computational Bio-Structural Model Explaining the Selective Killing Cancer Cells

The 3D structure of cardiolipin, the main mitochondrion outer membrane monomer, was reconstructed. Cardiolipin structural variations were determined according to the presence or absence of cytochrome c binding in normal and cancer cells, respectively. The binding between cardiolipin and HOO under all these conditions was analyzed. Cardiolipin bilayer was built using the Membrane Builder generator from the CHARMM-GUI web toolkit [20,21]. A bilayer was built using a deprotonate cardiolipin molecule in the same way. We minimized both these structures by performing a short molecular dynamic (MD) simulation using the software GROMACS [22]. For both systems, the pre-production minimization and relaxation protocols were automatically generated by the CHARMM-GUI Input Generator. They consisted of 5000 steps of energy minimization, keeping a constant volume and a temperature of 303.15 K (NVT) using the Berendsen thermostat for 20,000 steps with a 1 fs time step. The production runs adopted a time step of 2 fs. In this case, each MD was launched for total 1 ns of simulation.

### 2.6. Analysis of the Relationship between Ozonide Amount and Cancer Cell-Killing Effect

The relationship between ozonide amount and cancer cell-killing effect was examined by analyzing comparatively 9 ozonized oils having different levels of ozonides. The following formulations were evaluated by comparing them to the untreated control or to control treated with non-ozonized sunflower oil (sham-control): ≤100 ozonides (Ozone Elite oil, Ozone cream oil 10, Oil olive O3 TuPiel), ≤300 ozonides (VO3 active spray, Prog. Olive oil, EMI sunflower oil, Oil Ozofarm), ≥700 ozonides (HOO 700), ≥1100 ozonides (HOO 1100).

Anaplastic carcinoma cells A549 were grown in the presence of different ozonized oils, as previously reported; their ability to induce cell death was assessed by crystal-violet viability assay. For each formulation, the quantity of cells still vital after treatment (percentage as compared with the untreated control bearing 100% vitality) was evaluated. All the experiments were replicated 8 times for a total of 88 independent experimental analyses (11 experimental conditions × 8 replicates).

### 2.7. Inability to Kill Normal Cells

The inability of HOO to induce cytopathic effects in healthy cells (safety) was tested in primary differentiated human keratinocytes (Biological Bank and Cell Factory, IRCCS Policlinico San Martino, Genoa, Italy) treated for 1–3 h with HOO, 80% *v*/*v* with the culture medium. The results were examined at 48 and 72 h after treatment.

### 2.8. Synergism with Radiotherapy

This experiment was performed to solve the problem of HOO application-timing in relation to radiotherapy, i.e., whether a synergistic effect in killing cancer cells exists or not. In the case of positive answer, it should be clarified whether to apply HOO before or after the treatment with gamma radiation. To face these problems, an in vitro experiment was performed in anaplastic carcinoma cells (A549) exposed to ionizing radiation (2 Gy) undergoing HOO treatment either before or after radiation treatment. Cell survival was evaluated by crystal violet staining, and results obtained in OHOO-treated cells compared with those obtained in control cells exposed to radiation and treated with sunflower seed oil (sham-control). The experiment consisted of 16 replicates in multi-well plate for a total of 48 experiments (3 experimental conditions × 16 replicates).

### 2.9. Experimental Evidence in Human Subjects

The use of ozonized oil per os in human subjects as a food integrator was approved by the Health Ministry of Malta (approval number 0075/2020 according to EC1924/2006) issued on 17 March 2020.

Five healthy male subjects aged 49.2 ± 12.7 years old were treated for 1 week, administering 12 mL of HOO per os per day. Blood samples were collected before administration (T0) and after treatment (T1) and used for immunological analyses by FACS. The influence of HOO on blood monocytes was performed using HLAdr monocyte activation marker. NK and helper lymphocytes counts were performed using CD3 and CD4 markers. FACS analyses were performed using a LSR Fortessa X20 (Becton and Dickinson, Eysins, Switzerland).

A total of 115 cancer patients (age 58 ± 14 years; 44 males, 71 females) were treated with HOO contained in cellulose pills for 8 months as complementary therapy. The treatment was performed in parallel to standard chemo/radio therapeutic regimens performed for each cancer type according to the international guidelines.

The cancer diagnoses were the followings: brain (glioblastoma and astrocytoma) 22, pancreas adenocarcinoma 18, skin epithelioma (squamous and basal) 7, lung (NSCLC and small cell lung cancer) 12, colon adenocarcinoma 13, breast cancer (estrogen receptor positive) 24, prostate adenocarcinoma 7 (Gleason severity score >8), ovary and womb 5, kidney and bladder 5, non-Hodgkin’s skin lymphoma 2.

Cancer status at T0 (before HOO administration) and T1 (after HOO treatment) was examined by NMR, TAC, and PET, as performed for standard follow-up regimens according to international guidelines. Blood analyses were performed monthly. Amounts of oxidant (H_2_O_2_ milli-equivalent per 100 mL) and antioxidant (ascorbic acid milli-equivalent per 100 mL) in blood were examined monthly by the Free Radical Analysis System using a Fras4 Evolvo System (H&D, Parma, Italy) [23,24].

## 3. Results

### 3.1. Comparative Analysis of Various Ozonized Oils

A battery of oils having different ozonide content was tested for its ability to kill A549 cancer cells by evaluating the decrease in cell viability. The results obtained are shown in Figure 1.

Cancer cell viability was observed to be decreased in the presence of ozonized oils compared with controls. However, ozonized oils at low ozonide were unable to reduce the percentage of surviving cells to under 10%. The only formulations able to achieve this result were the HOOs. In fact, the percentage of surviving cells was only 6.8% in HOO 700 ozonide and reached the minimum value detected of 2.7% in HOO 1100 ozonide. The EC50 was calculated according to the exponential regression equation between ozonide and cell viability, obtaining a value of 433 ozonides. It is noteworthy that the dose–response effect observed for HOO depends on the amount of ozonide. This experiment showed that the amount of ozonide is the key element of the killing effect of ozonized oils against cancer cells.

Accordingly, HOO 1100 was selected for further analyses because of the highest level of cancer cell killing efficacy as compared with the other ozonized oils. Indeed, in HOO, ozonide content is equivalent to 800 meq O2/kg, corresponding to 220 mg of O3.

### 3.2. Experimental Evidence in Lung Cancer Cells

Human lung cancer A549 untreated cells and cells treated with sunflower oil (sham) rapidly grew and reached to confluency after 72 h. Conversely, A549 cells treated with HOO showed impaired growth during first 24 h. After this time, they rapidly underwent cell death, which culminated at 72 h. Cell death was characterized by (a) disappearance of the cell-growth carpet; (b) presence of dead cells in the supernatant; (c) diffuse apoptotic bodies. These results are shown in Figure 2. The lack of viability of cancer cells treated with HOO was also demonstrated at 24 h by Trypan blue staining selectively labelling only death cells (Figure 3).

### 3.3. Experimental Evidence in Glioblastoma Cancer Cells

After 12 h of HOO treatment, U87MG showed 4.68% of cell viability, as evaluated by MTT test, while A549 showed 8.43% of cell viability. After 24 h of treatment, U87MG showed 7.69% of cell viability and A549 showed 12.16% of cell viability. Accordingly, glioblastoma U87MG cells were more sensitive to HOO than lung adenocarcinoma A549 cells. This finding was well evident after 12 and 24 h of treatment. Regarding shorter time, there was no significant time difference between the two cell lines tested. Indeed, after 2 h the percentage of live cells was 13.91% for U87MG and 13.51% for A549; after 6 h of HOO treatment U87MG showed 6.23% of cell viability, while A549 showed 7.58%. These results are demonstrated in Figure 4.

### 3.4. Pharmacological Mechanism. Experimental Evidence in Lung Cancer Cells. Penetration inside Cell Cytoplasm

Sunflower seed oil and HOO penetration inside A549-treated cells was traced by microscope light scattering. Sunflower seed oil (sham-control) penetrated the cells (cytoplasm) only in a minimal amount and was compartmentalized (closed) into small well-defined vacuoles. Conversely, ozonized oil (HOO) penetrated abundantly into the cytoplasm, likely due to its peculiar ability to oxidize cell membranes, in particular the plasmatic membrane that delimits the cell from the external environment. Once penetrating the cytoplasm, the ozonized oil was initially compartmentalized into vacuoles; however, the membranes of these vacuoles were rapidly oxidized, and the oil spread into the cytoplasm overlapping intracellular membranes and organelles (Figure 5).

These events culminated after 24 h and were followed by death of cells treated with HOO.

### 3.5. Mitochondrial Status and Intracellular Calcium Release

The dose-dependent release of intra-cytoplasmic calcium was detected by rhodamine staining in cancer cells treated with HOO (Figure 6A). In the same cells, the mitochondrial membranes were selectively stained by green dye (DiOC6), verifying the decrease in signal intensity in HOO as compared with control cells treated with oil only (Figure 6B). This result reflected the damage of mitochondrial membranes, as induced by ozonized oil treatment. In cells receiving this treatment, it was also possible to observe the presence of large lipid vacuoles mainly located in the perinuclear zone (Figure 6, arrows). We carried out the characterization of these vacuoles by labeling them with LC3 in order to verify whether they were autophagic vacuoles, but the results were negative. Vacuole staining performed using dyes for lipid materials was not effective due to the rapid oxidative degradation of the dyes (Nile red) that was observed. Therefore, it is likely that these vacuoles are composed of oxidative lipids that cannot be catabolized by the cell. Because of this reason, the HOO-treated cancer cell takes the characteristic appearance of a ‘foamy cell’, which is characteristic of cells that accumulate oxidized lipids but cannot catabolize them due to the high level of mitochondrial damage. Indeed, lipid catabolism through the beta-oxidation biochemical pathway occurs inside mitochondria. The presence of these vacuoles in HOO-treated cells further demonstrates that the HOO action is expressed preferentially and directly towards the mitochondrial membranes.

In Figure 6C, fluorescence in cultured cells was evaluated by opening both channels (red and green) in order to check the overlay between mitochondrial membrane damage (attenuated green light) and extramitochondrial calcium release (red). In the case of overlap, the resulting color was yellow, otherwise the green and red colors were maintained.

In HOO-treated cancer cells, the images showed a yellow staining. This result indicates that the release of intracellular calcium (red) took place exactly from the mitochondrial membrane (green) damaged by HOO. In fact, this overlap (yellow color) did not exist in control cells where the mitochondrial membranes were not damaged, and there was no release of calcium from the mitochondria. Accordingly, the green color was maintained and no red color was shown. This result indicates that in cancer cells treated with sunflower oil (sham-control), calcium remained compartmentalized inside mitochondria; conversely, in HOO-treated cells, calcium was abundantly released from the mitochondria and spread into the cytoplasm. This result shows that the intracellular induction of mitochondrial damage with the consequent activation of intrinsic apoptosis was the main mechanism underlying the anticancer action of HOO.

### 3.6. Field Emission Scanning Electron-Microscope and EDX Analyses

Figure 7 shows the FE-SEM images of both the control and HOO-treated cells. At low magnification (a, e), the change of morphology induced by the ozonide treatment on the cells is evident. The HOO-treated cells underwent dramatic smoothing and rounding (b, f), disintegration and death (c, g), and the size decreased (d, h).

The number of elements contained in cells was estimated by EDX analysis. The carbon/oxygen ratio was high (C/O 4.3) in cancer cells, a situation envisaging the presence of the reducing intracellular environment characterizing these cells (left panel). Conversely, the carbon oxygen ratio became extremely low (C/O 1.5) in HOO-treated cancer cells, a situation envisaging the occurrence of oxidative stress in the intracellular environment as well as lipids and carbon-chain structure oxidation (right panel).

### 3.7. Evaluation of Apoptosis

Annexin V & Dead Cell Assay showed that apoptosis is the main mechanism of cell death affecting A549 cells treated with both HOO700 and 1100. Indeed, after two hours of treatment, control cells showed 11.10% of total apoptosis, with a clear prevalence of late apoptosis (40.90%) on early apoptosis (3.25%). Cells treated with HOO700 showed 30.35% of total apoptosis, divided into late apoptosis (29.10%) and early apoptosis (1.25%). HOO1100 determined 46.20% of total apoptosis (45% of late apoptosis and 1.20% of early apoptosis). These results are shown in the upper panels of Figure 8.

Regarding 4 h of treatment, in control cells, 9.30% of total apoptosis (7.35% of late apoptosis and 1.95% of early apoptosis) was detected, 50.80% in cells treated with HOO700 (46.95% late and 3.85% early apoptosis) and 47.65% in cells treated with HOO 1100 (45% late apoptosis and 2.65% of early apoptosis). These results are shown in the bottom panels of Figure 8.

### 3.8. Computational Bio-Structural Model Explaining the Selective Killing of HOO towards Cancer Cells

The results of the molecular dynamic simulations suggested two different conformations for the processed systems. In the case of active mitochondrion (healthy cell), the cardiolipin bilayer was thick, tight, and symmetrical, with no breaks of continuity between the hydrophilic heads protruding into the cytoplasm. This solid structure prevented the access to the hydrophobic tails of cardiolipin of the oxidizing radicals present in cytoplasm, such as those carried by OHOO and indicated by red circles in Figure 5. Therefore, the mitochondrion of the normal cell was resistant to the killing effects of OHOO. This structural situation is reported in Figure 9 left panels.

Conversely, in cancer cells, cardiolipin modified its structure due to the absence of the interaction with a functioning cytochrome c (the pivotal effector of aerobic glycolysis). Under these circumstances, the angle of convergence of the hydrophobic legs with the hydrophilic head was increased, a situation resulting in the divarication of the hydrophobic tails. This phenomenon occurred in all the cardiolipin monomers of the mitochondrial membrane, amplifying this molecular variation on the whole mitochondrial membrane. Thus, the mitochondrial membrane in cancer cell appeared at bioinformatic model thinner than in normal cells. This divarication of the hydrophobic tails created breaks of continuity between the hydrophilic heads of the cardiolipin that protrude into the cytoplasm, allowing access of HOO-oxidizing radicals to the hydrophobic tails of cardiolipin. According to this computational model, the mitochondrion of cancer cell was specifically sensitive to the damaging effects of HOO. This structural situation is reported in Figure 9 right panels.

### 3.9. Inability to Kill Normal Cells

Results indicate that neither alteration of cell viability nor cytopathic effects occurred in noncancer cells treated with HOO, as demonstrated in skin keratinocytes (Figure 10).

Cell viability of keratinocytes treated with ozonized oils quantified by MTT test was 100% in control cells, 99.8% after 1 h, 99.4% after 2 h, 98.7% after 3 h in ozonized oil-treated cells. Cell viability was not evaluated at times >3 h because the oil interface blocked cell exchange with culture medium, causing cell sufferance both in sham-treated cells (sunflower oil) and ozonized oil-treated cells, independent of oil toxicity.

### 3.10. Synergism with Radiotherapy

A strong radio-sensitizing effect of HOO 700 and even more of HOO 1100 was detected. The maximum effect observed was obtained treating A549 cancer cells with HOO 1100 after their exposure to gamma rays. From a mechanistic point of view, this effect was in line with the activation of the intrinsic mitochondrial apoptosis activated by HOO in cancer cells undergoing genotoxic damage induced by radiotherapy. The results obtained are shown in Figure 11.

A summary of the results obtained in vitro is reported in Table 1.

### 3.11. Experimental Evidence in Human Subjects. Healthy Subjects

Two volunteer healthy subjects (males, 55 years old) were treated with 12 mL of HOO 700 for 1 week twice per day away from meals. Cytofluorimetric analysis (FACS) was performed to evaluate macrophages and lymphocytes pro-inflammatory activation in the peripheral blood before (T0) and after treatment (T1). The results obtained showed a marked decrease in macrophage activation markers (HLAdr) in both subjects, while no variations were observed in the lymphocyte subpopulations responsible for the protective immunity. An example of the cytofluorimetric results obtained is reported in Figure 12. Cytofluorimetric analysis evaluated T-helper CD4+ and cytotoxic T lymphocytes CD8+ before and after ozonized oil treatment without observing any variation. These results provide evidence that HOO can induce anti-inflammatory effect without causing immuno-suppression.

Drug safety was also evaluated in vivo by analyzing the traditional blood chemistry parameters of the two volunteers treated: no alterations were found in the basal physiological state.

### 3.12. Experimental Evidence in Human Subjects. Cancer Patients

The transferability of the results obtained in vitro and in vivo in animals was initially tested in seven human patients affected by cancer. The results obtained are summarized as follows.

### 3.13. PATIENT 1. SKIN CANCER

The patient was a 93-year-old female with malignant spino-cellular epidermoidal carcinoma, confirmed histopathologically. The neoplasia was localized in the scalp in the parietal region and presented ab initio a character of extreme invasiveness and rapid progression. In fact, in only a few weeks, the cranial case was invaded with consequent parietal osteolysis. The neoplasia continued its progression rapidly, penetrating the skull and taking on the arachnoid. These data were revealed by computerized axial tomography (TAC). The neoplasia had considerable size in the outer part, thus covering the whole skull with conspicuous growth—not only endophytic (inside the skull) but also exophytic (protrusion of the neoplastic mass outside the skull).

The cancer was highly malignant, characterized by a high level of anaplasia, rapid progression, presence of neoplastic ulcers of significant size, high inflammation of the peri-lesioned margins, and total absence of repair by granulomatous tissue in the areas surrounding the neoplastic ulcer. The baseline clinical situation, as observed at the patient’s first visit, is shown in Figure 13.

Therefore, it was decided to use HOO 1100 in a post-treatment regimen—i.e., at the end of each radiotherapy session. Thus, at the end of the first radiotherapy session, OHOO 1100, gelled at 4 °C, was applied to the neoplastic lesion through a glass depositor, and the ulcer was coated with gauze and hydrophobic bandage.

The treatment continued for eight consecutive sessions with 2 days of interval between each one. OHOO medication was renewed at the end of each radiotherapy session. Therefore, OHOO was left to act in the pathological area for 48 h in the interval between radiotherapy sessions.

At the second session, the presence of yellowish exudate already present before the apposition of HOO was observed.

The treatment with radiotherapy and HOO 1100 continued for a total of eight sessions. Changes in the neoplastic lesion following treatment are shown in Figure 13.

Microscopic examination showed a strong size decrease in the cancer mass both in amplitude and in depth. Furthermore, the formation of a granulation repair tissue at the margins of the lesion was observed.

The radiotherapy was then suspended, and only topical treatment with HOO 1100 continued. Medication frequency was reduced to once per week because of the impossibility of the caregiver to bring the patient to the hospital more frequently. Despite the discontinuation of the radiation treatment, the cancer did not grow but further continued its regression, as shown in Figure 13.

These last results demonstrate the specific inhibitory effect of HOO 1100 against the tumor, even in absence of the radiation treatment. At the end of the follow-up, patients showed only a soft eschar of exudative material resulting from the colliquative necrosis of the neoplastic tissue. The eschar was not cleaned because the disappearance of the neoplastic mass left exposed the subarachnoid arteries, which were pulsating below the eschar itself. This result showed that the disappearance of the tumor mass occurred not only at the esophitic but also at the endophytic level. This result is remarkable because only a preparation characterized by a high bioavailability can be able to reach even the deepest areas of the tumor mass through a simple topical application.

In order to establish whether the topical application of OOAO had a systemic effect on the oxidative balance, we performed free radicals’ analysis in blood plasma before (T0) the beginning and end (T1) of HOO treatment using a Fras 4 system. At T0, the parameters were dramatically altered with a particularly low value of oxidizing species, i.e., 38 U Carr (normal range 250–280 U). In parallel, the antioxidants were strongly increased, with a value of 8318 U Cor (normal range 2200–2800 U). The antioxidant/oxidant balance (UCor/UCar ratio) was therefore 219 (normal range 7–10). These results show that the presence of a cancer mass characterized by large size implied the strong decrease in oxidation at a systemic level. At T1, the values were changed as follows: 201 U Carr, 7544 UCor, and the ratio UCor/UCar was 37. Therefore, an increase of 530% of the oxidative species was observed, a decrease of 13% of the antioxidants, a decrease of 583% of the ratio UCor/UCar. These values indicate that the regression of the neoplastic mass was related to the variation of the oxidative balance induced by treatment with OHOO.

At the end of the treatment the patient was in good health and no longer suffering or feverish. During treatment, no subjective or objective side effects related to OHOO treatment were observed.

### 3.14. PATIENT 2. SKIN CANCER

There was an 86-year-old female patient with a recurrent skin ulcer in the palmar region of the right forearm. The lesion was subjected to biopsy and histopathological analysis, and then it was classified as ulcerated metatypical nodular basocellular carcinoma; the lesion extended in depth, infiltrating the papillary and reticular derma; the presence of surrounding chronic inflammatory infiltrates was observed. Before treatment, the lesion was macroscopically characterized, as reported in Figure 14.

Therefore, the lesion was definitively removed surgically. During treatment, no subjective or objective side effects were observed.

### 3.15. PATIENT 3. PROSTATE CANCER

Subject was a 55-year-old male affected by prostate cancer, as confirmed by biopsy and histopathological examination. Staging and severity were very high, with this cancer being classified with a Gleason score 9 (max scale value 10). Cancer dimension: 3.5 cm. Imaging (ECT) detected a trend towards invasion of the prostate capsule, peri-prostatic adipose tissue, left seminal vesicle, and local lymph nodes. Relatively low PSA (4.5 ng/mL) was observed due to the high anaplastic behavior and the poor differentiation. Before surgical treatment, the patient was treated for 60 days with daily intra-rectal administration of ozonized oil (HOO 700) and oral administration of ozonized oil 12 mL twice per day.

At surgery, the field macroscopically appeared clear and without any evident inflammation or defragmentation of the cancer mass. A similar situation usually did not occur in such a high-grade malignancy characterized by high inflammation, severe infiltration of surrounding structures, fast growth, and extreme fragility of the cancer mass. Accordingly, surgery removed the whole cancer, as well as surrounding tissues, seminal vesicles, and lymph nodes. Microscope histopathological analysis did not detect any sign of inflammation in the cancer parenchyma or surrounding tissues. The presence of tumor-associated macrophages was not detected at all, at variance with the typical aspect of this high-malignancy cancer (Figure 15).

Only 1 local lymph node, out of the 10 examined, was barely affected by cancer invasion.

Patient subsequently underwent anti-hormone therapy and radiotherapy as scheduled by the standard treatment protocol. In parallel, he continued ozonized oil treatment. After 6 months, no sign of relapse was detected.

The analysis of oxidative status in blood plasma indicated that oxidative stress was low before ozonized oil treatment (T0) (210 U Car, normal range 250–280) as well as the antioxidant/oxidant ratio (9.9 Ucor/UCar). After treatment, oil treatment (T1) oxidative stress was increased (295 U Car, 7.9 Ucor/UCar ratio). No adverse effect related to the ozonized oil treatment was detected.

### 3.16. PATIENT 4. PROSTATE CANCER

There was a 76-year-old male, affected by prostate cancer (adenocarcinoma), as confirmed by biopsy and histopathological examination (September 2015). No local or distant metastasis was detected. NMR revealed high metabolic rate, fast cell proliferation, and blood vessel proliferation. Biopsy detected severe inflammation of cancer mass and surrounding tissue. Intermediate malignancy: Gleason score was 6, and cancer dimension was 3.0 cm.

No results were observed in anti-hormonal therapy (November–December 2015). Standard radiotherapy regimen was of 40 sessions (January–March 2016). In October 2016, after radiotherapy, NMR detected cancer persistence with cell proliferation and remarkable blood vessel proliferation.

From January 2017, HOO oral treatment (12.5 mL twice a day for 18 months) was started. In February 2018, TAC and radioimaging demonstrated complete cancer disappearance (Figure 16).

No relapses were insofar detected (ongoing follow-up, 18 months).

No adverse effect related to the ozonized oil treatment was detected.

### 3.17. PATIENT 5. PROSTATE CANCER

There was a 74-year-old male affected by prostate cancer (adenocarcinoma), as confirmed by biopsy and histopathological examination. No metastasis was detected. High malignancy grade (Gleason 8), PSA 9.1 ng/mL was observed. Treatment with ozonized oil (intra-rectal once per day) and ozonized oral oil (12.5 mL twice a day) for 40 days before therapeutic surgery was started. At surgery, despite the high cancer malignancy, no invasion of prostatic capsule, surrounding adipose tissue, seminal vesicle, or lymph node was detected; blood vessel proliferation was observed. Full eradication was observed with no relapses after follow-up of 8 months.

### 3.18. PATIENT 6. BRAIN GLIOBLASTOMA

Female subject was a 3-year-old glioblastoma patient, as detected by TAC/NMR and confirmed by biopsy. High malignant grade was observed. Patient underwent standard radio/chemotherapy regimen. Ozonized oil was administered (12.5 mL twice a day for 90 days). The clinical follow-up was compared with those of three other young patients in a similar clinical situation but devoid of ozonized oil treatment. All of these three patients underwent fast cancer progression, and one died. Conversely, the ozonized oil led to the arrest of cancer growth and a dramatic decrease in cancer dimension that after 3 months of treatment was only 35% of that initially detected. A similar finding was totally unexpected in such a fast growing cancer.

Cerebrospinal fluid was collected, and oxidative status was analyzed. The sample was contaminated by red blood cells. For these reasons, the analysis of oxidative status was unreliable. Conversely, after several efforts, we were successful in analyzing the antioxidant status, whose values were 892 and 895 (replicate analyses on 15 uL × 2) U cor (umol/L ascorbic acid equivalent). The normal reference value was 2500, with a b max–min range of 1800, which was below the threshold. Accordingly, the observed value was extremely low (despite the red blood cell contamination releasing antioxidant). It could be concluded that the depletion of antioxidant due to the therapeutically induced oxidative stress (ozonized oil treatment) had been effective on the CSF of this patient. This value (893 U cor) could be assumed as a threshold to be reached to obtain therapeutic effects.

It is conceivable that ozonized oil, due to its high lipophilicity, is able to cross the blood–brain barrier. Specific in vitro and in vivo tests are ongoing to further substantiate this issue.

### 3.19. PATIENT 7. BRAIN GLIOBLASTOMA

Female subject was a 38-year-old affected by brain glioblastoma in left hemisphere (diagnosis July 2014. 1st NMR, (T0)). First surgery was performed in September 2014. High malignancy (grade III) was observed. In May 2017, relapses (2nd NMR) were detected, and in June 2017, there was a second surgery. September 2017 radiotherapy (60 Gy) in parallel with the start of ozonized oil therapy (oral administration, 6 mL per day) was performed. No relapse (September 2018) was detected (3rd NMR, T1). Cancer presence at T0 as well as its clearance at T1 are reported in Figure 17.

After these 7 patients, a total of 115 cancer patients, 76 males and 39 females, average age 60.1 ± 17.8 years, were treated with ozonized oil. An overview of the results obtained with regard to oxidative status in cancer patients treated with HOO is reported in Figure 18. For reference, oxidative status was also analyzed in parallel in 40 cancer-free subjects (22 males, 18 females, average age 58.0 ± 6.4 years).

Standard follow-up exams included hematological analyses, blood analysis of cancer markers (e.g., Ca-19), nuclear magnetic resonance, computerized tomography with and without glucose tracer, echography. Clinical outcomes observed in HOO-treated cancer patients as compared with cancer patients undergoing standard therapeutic regimens is reported in Figure 19, referring to all cancers. Data on clinical outcomes after treatment for each cancer type are reported in Table 2. Clinical outcomes were significantly different (chi-square *p* value < 0.0001) in cancer patients receiving standard treatment only as compared with those additionally receiving HOO as integrative therapy, as evaluated by chi-square test (Figure 19).

## 4. Discussion

Obtained results provide evidence that HOO is a new and interesting strategy for prevention of cancer progression and relapses. Indeed, patients undergoing HOO integration, in addition to standard therapeutic regimens, showed increased survival and decreased rate of relapses. These clinical outcomes are justified by the mechanisms analyzed in vitro to shed light on the effects of HOO in cancer cells. Cancer cells are highly sensitive to HOO oxidative effects due to their mitochondrial status. The oxidation of mitochondrial membranes can re-activate apoptosis in cancer cells [2]. The selective effect of HOO in killing cancer cells only without damaging normal cells is due to the different mitochondrial situation. The mitochondrion is inactive in a cancer cell, both as far as concerns metabolic (aerobic glycolysis) and pro-apoptotic function, that in normal cells is activated by the release of cytochrome c and calcium from the mitochondrion into the cytoplasm. This situation critically differentiates cancer from healthy tissue, explaining why cancer cells cannot die because of apoptosis while normal cells can. The outer mitochondrial membrane is predominantly composed of phospholipids, among which the most relevant is cardiolipin. This molecule is characterized by the presence of a hydrophilic phosphorylated head and two hydrophobic tails. Cardiolipin is organized to form the typical phospholipidic double-layer. However, the shape of cardiolipin is profoundly modified in relation to its binding to cytochrome c, a fundamental component of oxidative phosphorylation [26]. Cardiolipin displays its physiological structure only when bound to functioning cytochrome c or, in other words, in the active mitochondrion characterizing normal cells but not in cancer cells. The sensitivity to HOO is even higher for cancer stem cells that are antioxidant addicted. These cells have been selected among cancer cell pools because of their high level of antioxidant-based detoxifying mechanisms, allowing them to counteract the therapeutic effects of chemo/radiotherapies. The scavenging of these antioxidants that is exerted by HOO is an effective tool for making cancer stem cells sensitive to chemo/radiotherapies and for overcoming their resistance. This situation explains the improved clinical efficacy of standard therapeutic regimens in patients receiving HOO. This result may be achieved only by using ozonized oils having very high ozonide content, extremely pure, devoid of antioxidants, and highly bioavailable.

HOO exerts anticancer effects by activating various protective mechanisms including (a) scavenging of antioxidants from cancer cells; (b) re-activation of intrinsic apoptosis; (c) inhibition of the activation of tumor associated macrophages; (d) increase of oxygen tissue availability decreasing angiogenesis and metastasis.

An additional mechanism could be the competition with the mitochondrial fat oxidation metabolic pathway providing energy availability in cancer cells [27]. HOO is a metabolite of this pathway, but its catabolization inside the mitochondrion results in oxidative stress triggering mitochondrial damage, intracellular calcium release, and apoptosis in cancer cells. Targeting the intrinsic apoptotic pathway has been recently proposed as a new strategy against cancer [28].

Cancer is a systemic disease. The neoplastic mass is not able to develop autonomously in the absence of the trophic support provided by neo-angiogenesis and inflammation. The presence of a conspicuous infiltrate of inflammatory macrophage cells is a specific characteristic of malignant cancer. These macrophage cells are unable to counteract the neoplastic growth, supporting it by supplying oxygen, metabolites, and neo-vessels. These cells are referred as tumor-associated macrophages. The degree of inflammation is one of the most predictive and prognostic indexes of the unfavorable development of a neoplasm. Therefore, cancer should be counteracted not only at a topical level but also at the systemic level, controlling inflammation and oxidative status in the whole organism. In this regard, HOO could represent a significant step forward. HOO kills cancer by targeting cancer stem cells and activating apoptosis but also by exerting anti-inflammatory effects at the systemic level. Herein, presented results indicate that HOO induces in vivo anti-inflammatory effects without causing immuno-suppression. The mechanism underlying this situation is the inhibition of macrophage oxidative burst. Activated macrophages release oxygen-reactive species and inflammatory cytokines in the tissue environment in order to neutralize bacteria, if present. However, in the absence of bacteria, macrophage activation leads to an inflammatory response, which can assume pathogenic relevance by promoting cancer growth and progression. HOO inhibits macrophage oxidative burst through a negative feedback mechanism; indeed, the presence of an extra-cellular environment enriched with ozone-oxidizing species blocks the release of further oxidizing species from the macrophages, thus inhibiting their activation and the consequent inflammation.

The trophic support to solid cancers is provided by neoangiogenesis. This process is activated by low oxygen availability in cancer tissue, triggering hypoxia inducible factor representing the main activator of vascular growth factor release and blood vessel formation [29]. Herein, presented results provide evidence that HOO effectively releases oxygen species inside cancer tissue, thus counteracting the hypoxic situation triggering neoangiogenesis.

## 5. Conclusions

In conclusion, the experimental studies performed provide evidence of the efficacy of HOO treatment in killing cancer cells, thus integrating and potentiating the therapeutic effects of standard therapies. This conclusion is supported by the biological plausibility of the specific mechanisms activated by HOO in cancer cells, mainly including mitochondrial damage and activation of apoptosis. These effects are exerted by HOO due to its peculiar characteristics including: (a) oxidant effectiveness due to the high level of ozonide content; (b) dose customization by evaluating oxidant/antioxidant balance in blood plasma as well as cancer stage; (c) anti-inflammatory effects; (d) increase in oxygen tissue availability, counteracting cancer metastasis triggered by local hypoxia; (d) capacity of penetration inside cancer cells.

These findings provide evidence that HOO is endowed with a potential therapeutic efficacy against cancer in the absence of detectable side effects. Due to its pharmacokinetic and pharmacodynamic peculiarities, HOO represents an innovation in the field of complementary cancer therapy worthy of further clinical studies. Our result provides evidence that oral administration of ozonized oils with high ozonide content is a novel strategy for the prevention of cancer relapses and chemo/radioresistance. This approach could be used in clinical practice to fulfill the lack of anticancer treatments occurring in intervals between chemo/radio therapeutic regimens.

## Figures and Tables

**Figure 1 cancers-14-01174-f001:**
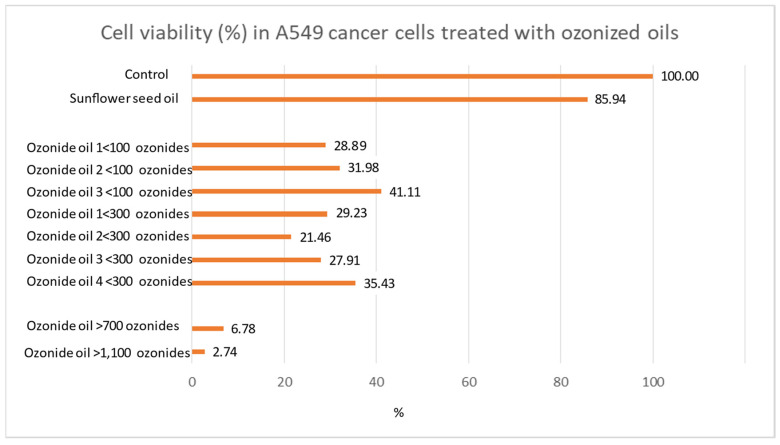
Comparative evaluation of ozonized oil effect on cancer cell viability (MTT test). Only ozonized oils having an ozonide content >700 decrease cancer cell viability below 10% in 24 h.

**Figure 2 cancers-14-01174-f002:**
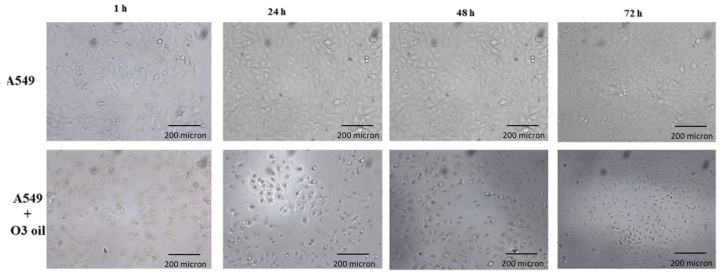
Time-dependent in vitro growth of A549 lung cancer cells either sham-treated with sunflower oil (upper row) or treated with ozonide oil >700 ozonides with a percentage *v*/*v* of ozonized oil related to sunflower of 97% (lower row). Sham-treated cells grow rapidly reaching semi-confluence at 24 h and full confluence at 48 h. Cancer cells treated with ozonide oil already display impaired growth during the first 24 h. After this time, they rapidly undergo time-related increasing cell death, culminating at 72 h.

**Figure 3 cancers-14-01174-f003:**
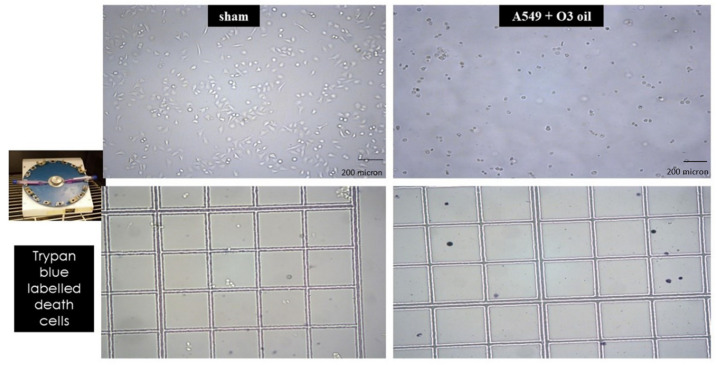
Detection of non-viable cells by Trypan blue staining after 24 h since seeding of A549 cancer cells, either sham-treated with sunflower seed oil (left panels) or treated with ozonide oil >700 ozonides (right panels). Upper panels, standard microscopy; lower panels microscopy after Trypan blue staining. Death cells are detected only in ozonide oil treatment.

**Figure 4 cancers-14-01174-f004:**
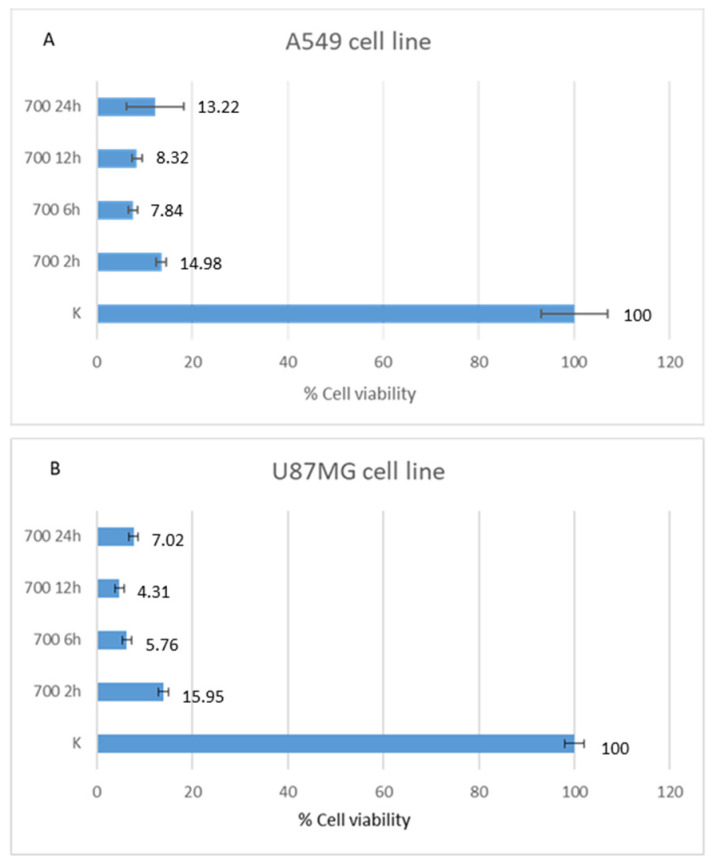
Viability comparison between human lung adenocarcinoma (A549, **A**) and glioblastoma (U87MG, **B**) cell lines treated with HOO for different times (2, 6, 12, and 24 h).

**Figure 5 cancers-14-01174-f005:**
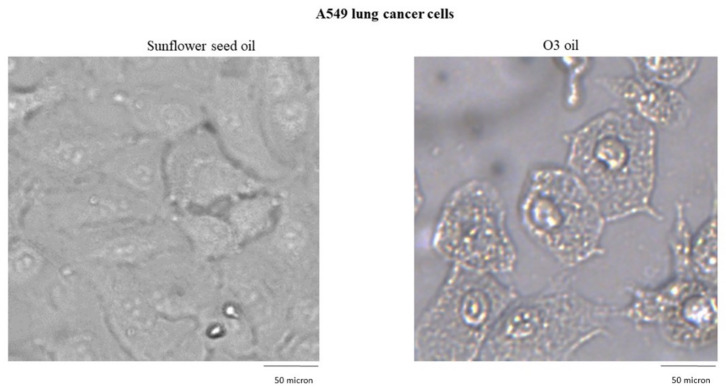
HOO penetration inside A549 lung cancer cells, as traced by microscope light scattering.

**Figure 6 cancers-14-01174-f006:**
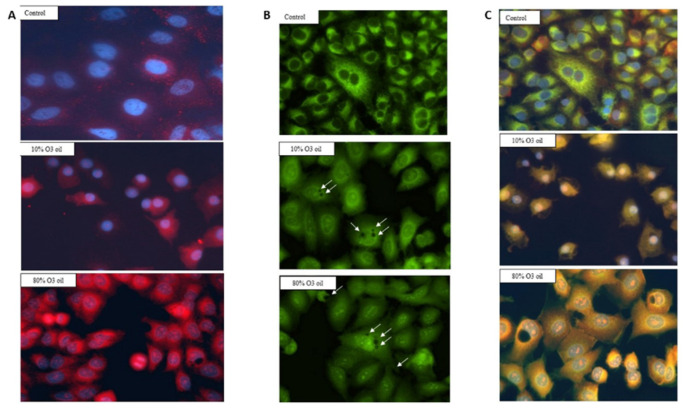
Intracellular calcium release as induced by ozonized oil in A549 lung cancer cells detected by rhodamine staining (red) and fluorescence microscopy. Mitochondrial membranes are labeled by DiOC6 staining (green). Red and green overlap reading both channels at the same time (right columns) indicates that calcium is released from mitochondria (yellow). Nuclei are colored by DAPI staining (blue).

**Figure 7 cancers-14-01174-f007:**
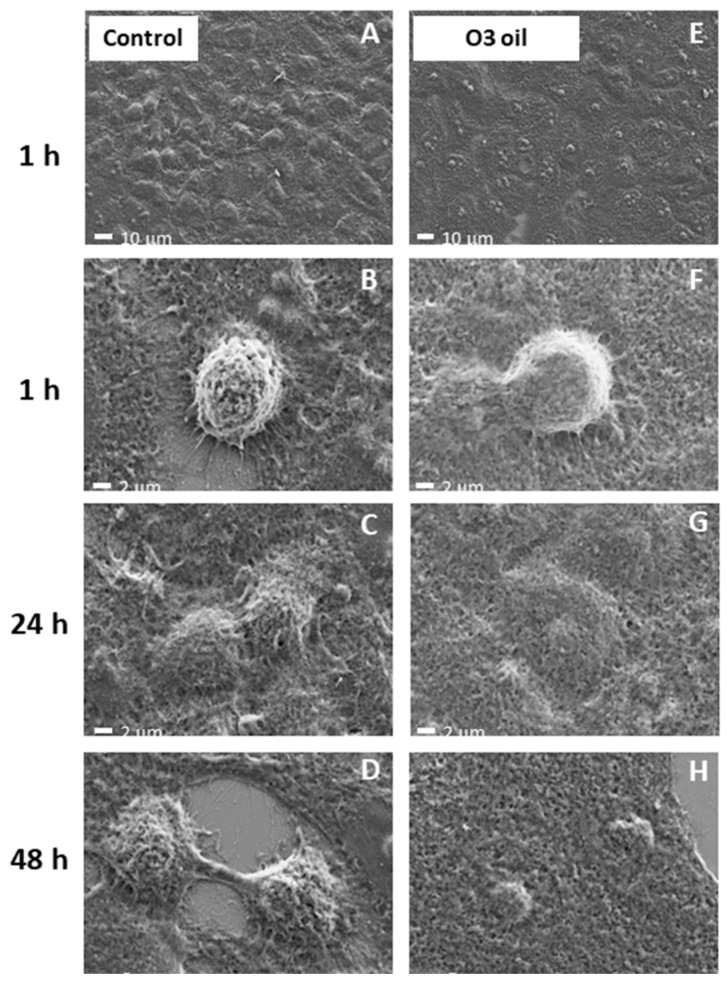
SEM images of the cells before (**A**–**D**) and after the ozonide oil (>700 ozonides) treatment (**E**–**H**) at a magnification of ×2000 (**A**,**E**) and ×10,000.

**Figure 8 cancers-14-01174-f008:**
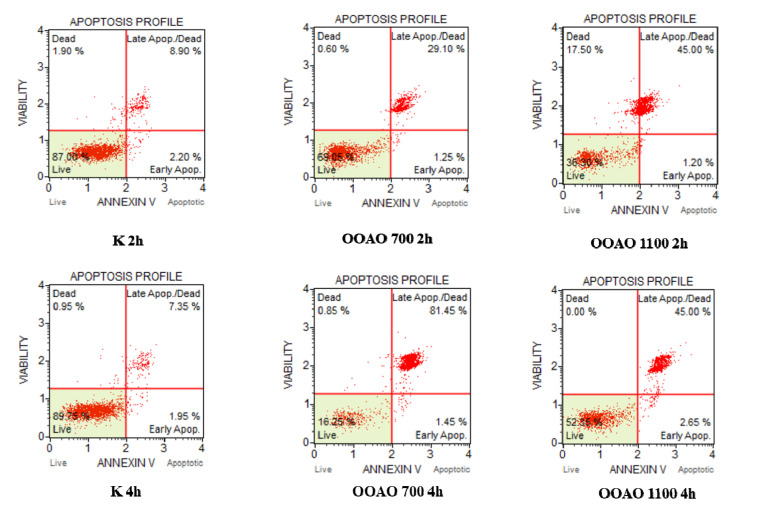
Apoptosis profile (Muse™ Annexin V & Dead Cell Assay) for A549 cells treated with 10% *v*/*v* of OOAO 700 and OOAO 1100 and untreated cells (K). Profiles were determined 2 h (upper panels) and 4 h (bottom panels) after treatment. Each plot has 4 quadrant markers, reflecting the different cellular states: the upper left quadrant contains dead cells (necrosis), the upper right has late apoptotic/dead cells (cells that are positive both for Annexin V and for cell death marker 7-AAD, 7-Aminoactinomycin D), the lower left contains live cells, and the lower right early apoptotic cells (cells that are positive only for Annexin V).

**Figure 9 cancers-14-01174-f009:**
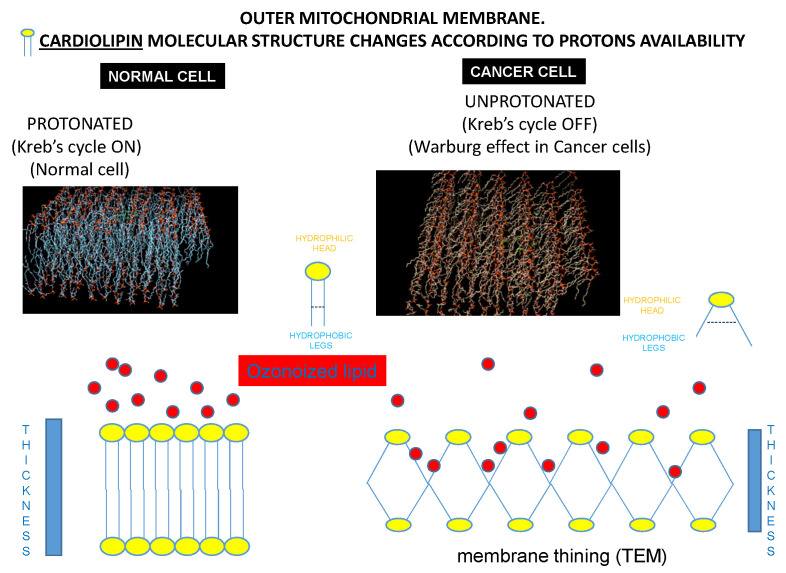
Bioinformatic analysis of mitochondrial membrane in normal (left panels) and cancer (right panels) cells. Mitochondrial membrane monomer cardiolipin undergoes a 10 A increase in distance of hydrophobic tails when not bound with cytochrome c. This situation occurs in cancer cells, resulting in mitochondrial membrane thinning. Accordingly, only in cancer cells can ozonized lipid (red circles) reach the hydrophobic legs of cardiolipin, thus causing mitochondrial membrane damage. This situation does not occur in normal cells, where ozonized lipid cannot reach this molecular target.

**Figure 10 cancers-14-01174-f010:**
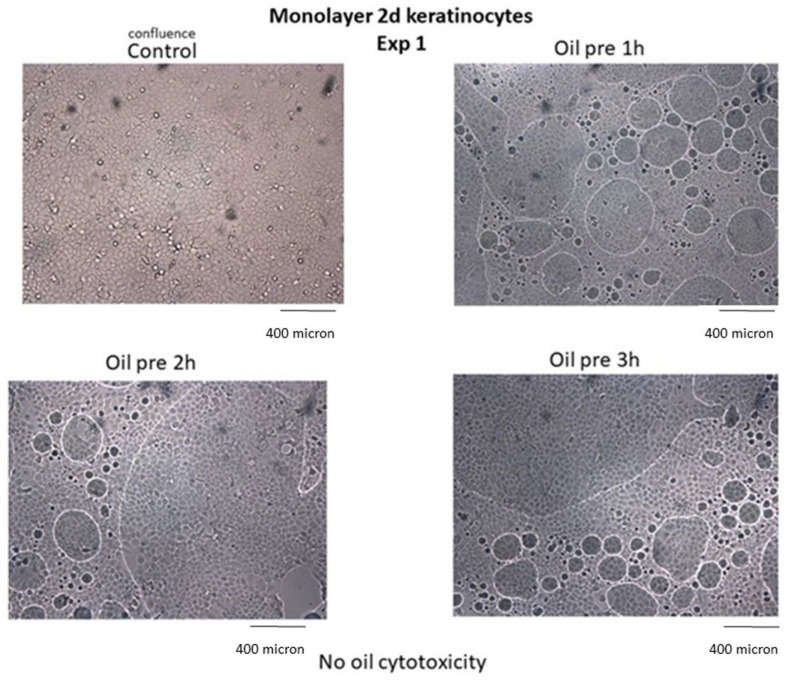
Ozonized oil (bubble oil in the culture medium 1, 2, 3 h) does not induce alterations in normal human keratinocytes. No change in cell viability, intercellular adhesion, and substrate adherence occur in ozonized oil-treated cells (Oil) as compared with untreated control.

**Figure 11 cancers-14-01174-f011:**
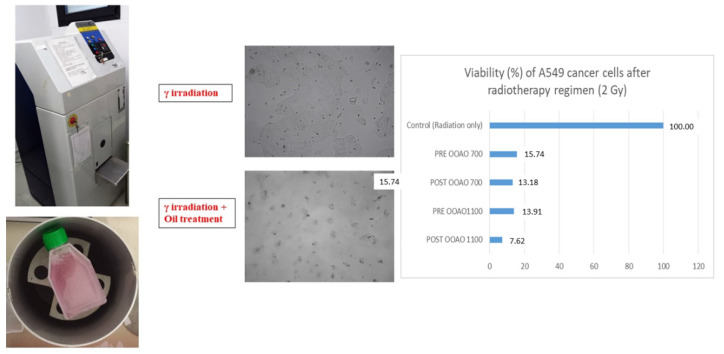
Synergism between gamma ray radiation and ozonized oils (OOAO) treatments in killing lung cancer cells.

**Figure 12 cancers-14-01174-f012:**
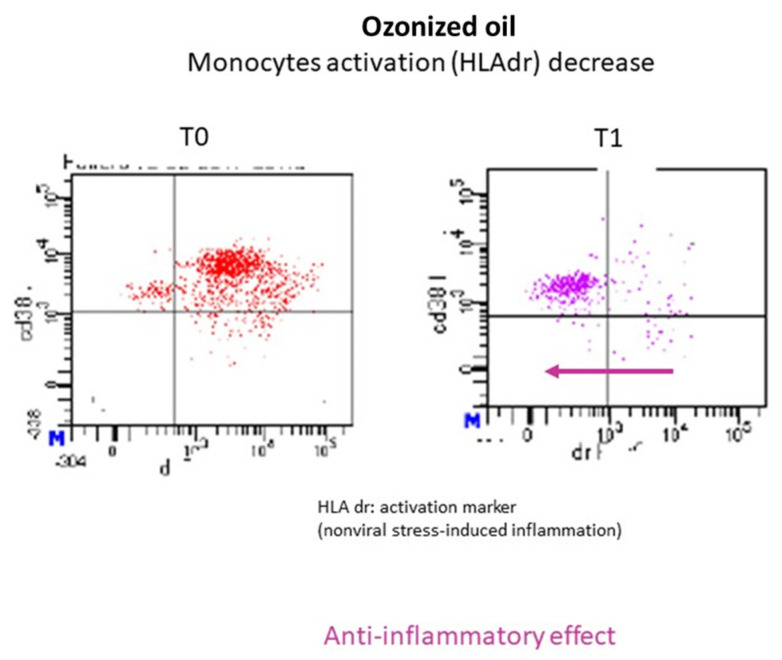
FACS analysis of blood monocytes (cd38, vertical axis). Activated monocytes are detected by analyzing HLAdr (horizontal axis). The number of HLAdr-positive activated monocytes was decreased (arrow) after 1 week of HOO treatment (T1, right panel) as compared with those detected in the same subject before treatment beginning (T0, left panel).

**Figure 13 cancers-14-01174-f013:**
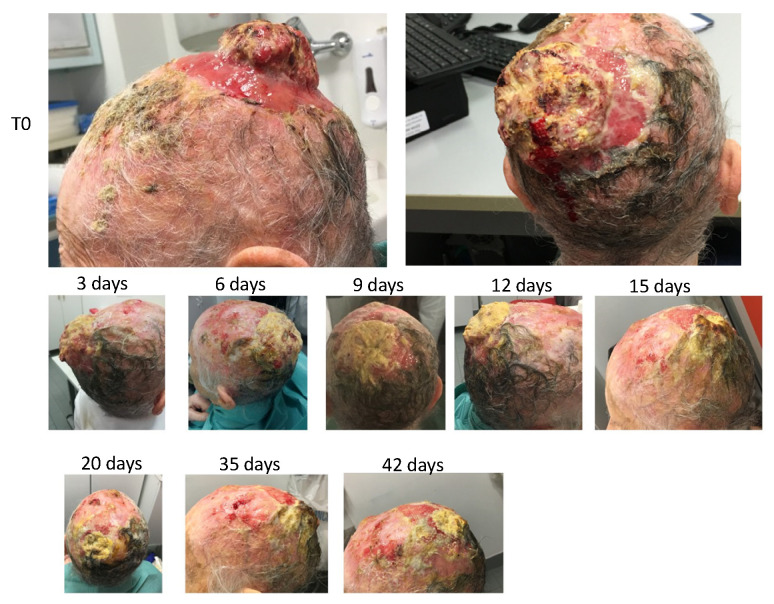
Evolution of a radioresistant skin basal carcinoma in a female 93-year-old patient treated with HOO.

**Figure 14 cancers-14-01174-f014:**
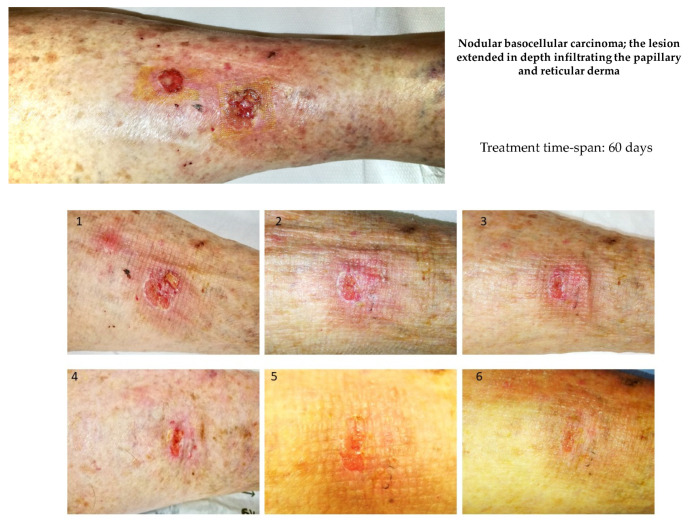
Evolution of an ulcerated nodular basocellular carcinoma in a female 86-year-old patient. The lesion extended in depth, infiltrating the papillary and reticular derma; the presence of surrounding chronic inflammatory infiltrates was observed before treatment (T0). HOO treatment induced cancer regression and disappearance of the surrounding inflammatory halo after 60 days.

**Figure 15 cancers-14-01174-f015:**
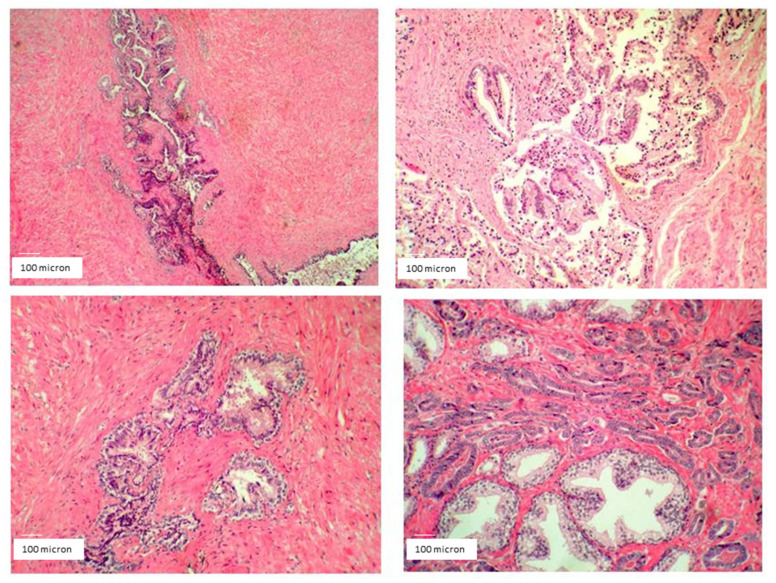
Prostate cancer histopathological analysis detecting poor signs of inflammation in the cancer parenchyma and surrounding tissues despite the high malignancy grade (Gleason 9) after HOO treatment. The presence of tumor-associated macrophages was not detected at all, at variance with the usual aspect of this cancer type (magnification 100×).

**Figure 16 cancers-14-01174-f016:**
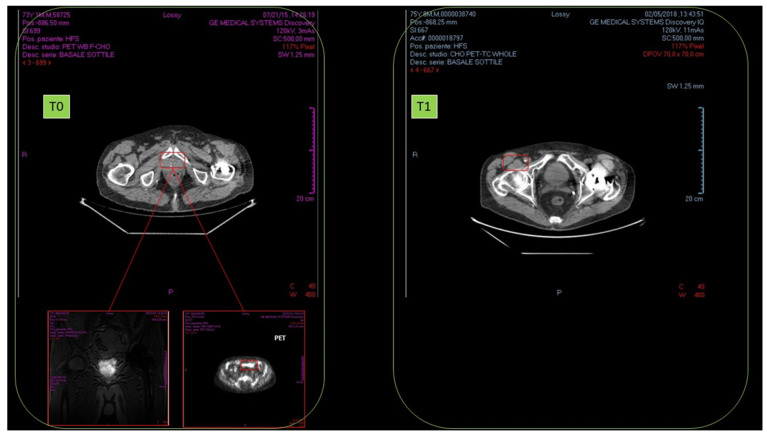
Disappearance of radioresistant prostate carcinoma (T0) after 1 year of HOO treatment (T1), as detected by nuclear magnetic resonance in a 76-year-old male patient.

**Figure 17 cancers-14-01174-f017:**
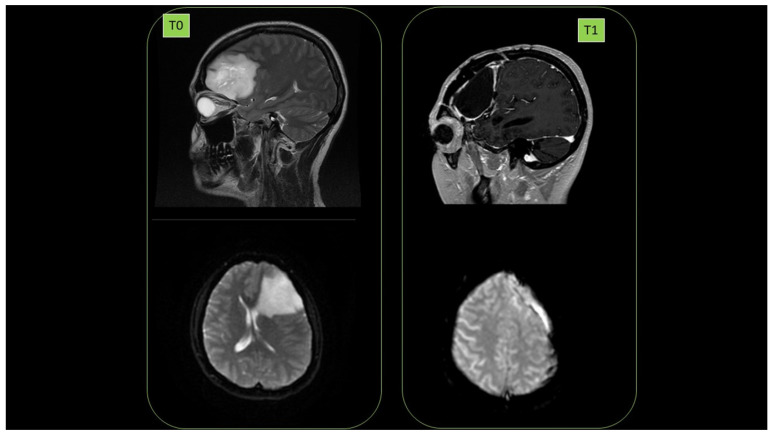
Disappearance of radioresistant brain glioblastoma (T0) after 1 year of HOO treatment (T1), as detected by nuclear magnetic resonance in a 38-year-old female patient.

**Figure 18 cancers-14-01174-f018:**
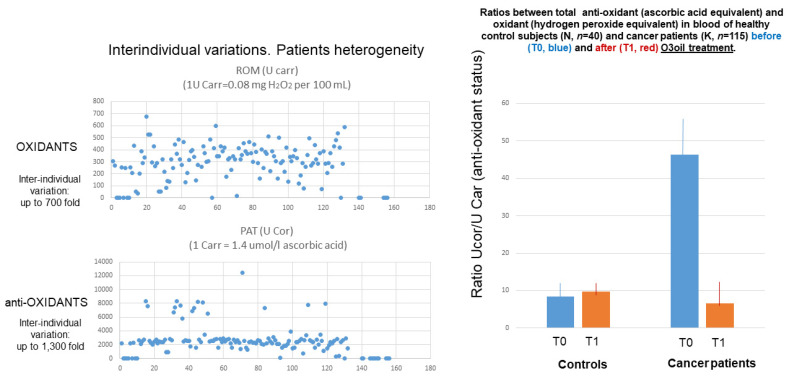
Analysis of oxidative status in the blood plasma of 115 cancer patients. The level of antioxidant was higher in cancer patients as compared with controls (left panel, blue columns). HOO treatment in cancer patients decreased the high level of antioxidant (left panel, red columns) moving back their amount to the level of unaffected controls.

**Figure 19 cancers-14-01174-f019:**
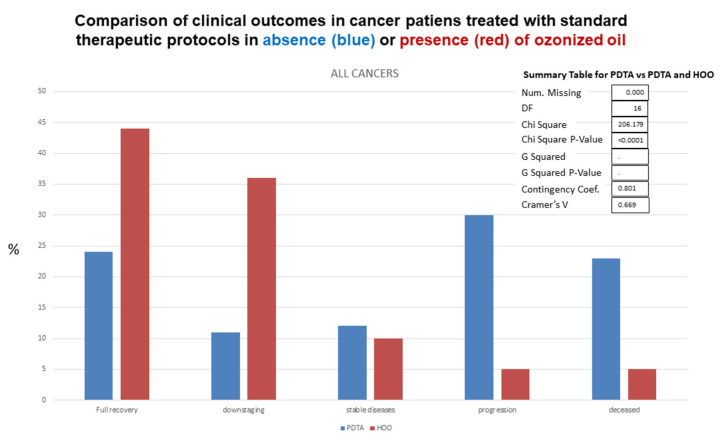
Clinical outcomes observed in HOO-treated cancer patients (red columns) as compared with cancer patients undergoing standard therapeutic regimens (blue columns), referring to all cancers.

**Table 1 cancers-14-01174-t001:** Summary of results from in vitro experiments dealing the effect of ozonized oil at high ozonides (HOO) in cancer cells.

Test	Method	Results	Reference Figure
Lung cancer cells killing	A549 cells, crystal violet staining, MTT test	Time-dependent cell death	Figure 2
Glioblastoma cancer cells killing	U87MG cells, crystal violet staining	Time-dependent cell death. Higher sensitivity than A549 cells	Figure 4
Alteration of mitochondrial membranes	A549 cells, fluorescence microscopy, bio-informatic analyses	Specific sensitivity to oxidation of cancer cells, calcium release	Figure 6 Figure 9
Cell morphology alteration	A549 cells, scanning electron microscope	Decreased size, rounding, detachment from support	Figure 2 Figure 3 Figure 5 Figure 7
Oxidation of intracellular organic carbon	A549 cells, X-ray diffraction	Oxidation of intracellular organic carbon	Figure 7
Apoptosis	A549 cells, annexin V labeling, FACS	Increase in apoptosis	Figure 8
Inhibition of macrophage activation	Raw264 cells, Lps activation, microscopy	No change in macrophage morphology, maintenance of rounding morphology, no emission pf pseudopods	[25]
Inability to kill normal cells	Primary differentiated human keratinocytes	No detachment from support, no growth inhibition	Figure 10
Synergism with gamma radiation in cancer cells killing	A549 cells, gamma radiation, crystal violet staining	Increased cell apoptosis and necrosis	Figure 11

**Table 2 cancers-14-01174-t002:** Clinical outcomes in cancer patients treated with HOO (follow-up 4 years).

Cancer Type	Full Recovery	Downstaging	Stable Diseases	Progression	Deceased	Total
Brain glioblastoma	32% (*n* = 7)	41% (*n* = 9)	9% (*n* = 2)	5% (*n* = 1)	13% (*n* = 3 *)	*n* = 22
Breast adenocarcinoma	67% (*n* = 16)	20% (*n* = 5)	5% (*n* = 1)	0% (*n* = 0)	8% (*n* = 2 *)	*n* = 24
Colon adenocarcinoma	54% (*n* = 7)	23% (*n* = 3)	8% (*n* = 1)	8% (*n* = 1)	8% (*n* = 1 *)	*n* = 13
Kidney/bladder cancer	60% (*n* = 3)	20% (*n* = 1)	0%	0%	20% (*n* = 1 *)	*n* = 5
Non-Hodgkin’s skin lymphoma	50% (*n* = 1)	50% (*n* = 1)	0%	0%	0%	*n* = 2
Lung NSCLC 7	14% (*n* = 1)	43% (*n* = 3)	29% (*n* = 2)	14% (*n* = 1)	0	*n* = 7
Lung SCLC 5	0% (*n* = 0)	60% (*n* = 3)	20% (*n* = 1)	20% (*n* = 1)	0%	*n* = 5
Ovarian/womb cancer	40% (*n* = 2)	40% (*n* = 2)	20% (*n* = 1)	0% (*n* = 0)	0% (*n* = 0)	*n* = 5
Pancreas adenocarcinoma	39% (*n* = 7)	44% (*n* = 8)	11% (*n* = 2)	6% (*n* = 1)	0% (*n* = 0)	*n* = 18
Prostate cancer	58% (*n* = 4)	28% (*n* = 2)	14% (*n* = 1)	0% (*n* = 0)	0% (*n* = 0)	*n* = 7
Skin cancer (radioresistant epidermoidal and basal carcinoma)	28% (*n* = 2)	44% (*n* = 3)	14% (*n* = 1)	14% (*n* = 1)	0% (*n* = 0)	*n* = 7
Grand total	*n* = 50	*n* = 40	*n* = 12	*n* = 6	*n* = 7	*n* = 115
Average	44%	36%	10%	5%	5%	

* Cancer stage at recruitment T4-N3-M1.

## Data Availability

All data are available upon request to the corresponding author.

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
