# Peer review of "Efficacy of High-Ozonide Oil in Prevention of Cancer Relapses Mechanisms and Clinical Evidence"

_cancers, 2022, doi:10.3390/cancers14051174_

Round 1
Reviewer 1 Report
This is quiet an interesting study with interesting and significant results that were presented.
Although the study have several points for criticism, we shall avoid these issues and concentrate on the main findings that current study presented and a lot of bright side of the study. Introduction is well presented. Material and methods section is pointing to the more or less good methodology used. Results are clearly presented and discussed very well.
Regarding conclusion section, based on your experience, what do you suggest that should be implemented in clinical practice and how?
After correcting technical issues throughout the paper, I suggest its publication.
Author Response
Dear Editors,
We would like to thank you for considering the manuscript entitled “Efficacy of high-ozonide oil in prevention of cancer relapses. Mechanisms and clinical evidences” by Izzotti A. et al., and for sharing the Reviewers’ comments that certainly helped in improving the quality of the manuscript (cancers-1573983). We appreciated the Reviewers’ comments, and we revised the manuscript accordingly. Please find enclosed to the submission of the revised version of the manuscript the point-by point reply to the Reviewers’ comments. For clarity’s sake, changes in the revised MS are marked in yellow.
We hope that the revised version of our MS will be now suitable for publication in the Cancers.
Accordingly, we prepared a revised version of the manuscript acknowledging Referees’ and Editor’s comments as below specified:
Reviewer 1:
COMMENT 1. Regarding conclusion section, based on your experience, what do you suggest that should be implemented in clinical practice and how?
ANSWER 1. A sentence reporting what our results suggest should be implemented in clinical practice has been added in Conclusions (last paragraph).
Reviewer 2 Report
I want to compliment the authors about this manuscript; really nice work—especially the clinical data about the ozonide use in patients. The authors can find some suggestions and questions down below.
line 226-230: To clarify cytotoxicity the EC50 value would be important to calculate.
Figure 2: What is the content/"concentration" between the sunflower oil and ozonide oil so that we can compare directly. I am assuming that at least a dilution of sunflower oil was done. It should be added that information.
Figure 3: Trypan blue not only detect death cells but all the non-viable cells. Change it on the figure and on the text.
Figure 4: Please, add values to this graph to be easily compared with figure 11
Figure 6: This merged definition needs to be clarified. It is missing one channel to show before merging one. It seems that the authors do not always present the same field to compare different staining (B and C). Missing more information on this figure.
Timelines: I do not understand why we have so many different timelines according to the assay. As an example, Figure 1 MTS was done at 2h, 6h, 12, 24h. So when apoptosis is studied, why do not choose the same timelines, instead, a new 4h is added. Can you please justify these different timelines, different choices in all the manuscript?
line 405: How can the author conclude that there is no cell viability alteration on up to 3h incubation? How did the authors conclude there is no cytotoxicity? Did they let the cell grow and stay more than 3h with the tested oil? Was any quantification done or just observed in a light microscope?
Table 1: not very comfortable with some conclusions. For example mitotic arrest (how was it checked?) and keratinocytes-related conclusions. This table is very useful, but I suggest adding the figure to the corresponding conclusion to be easier to follow this resume table. Double-check if the study supports all your resume conclusion in this table.
Line 433, 434: "These results provide evidence that 433 HOO can induce anti-inflammatory effect without causing immunosuppression." To be clear to the reader your conclusion, you should present cell subset number/percentage in the blood analysis that was done (supplementary figure).
Metabolics analysis would be much more advantageous to understand how mitochondria were affected or which pathway could be affected with ozonide.
There is no scale on the microscope figures.
Amazing in vivo results! But on these follow-up patients, can you clarify which kind of follow up exams were done. Do they develop any different comorbidities? This data will be important if you have it to add as supplementary information.
Author Response
Reviewer 2:
COMMENT 1 line 226-230: To clarify cytotoxicity the EC50 value would be important to calculate.
ANSWER 1 A sentence reporting the EC50 has been added.
COMMENT 2. Figure 2: What is the content/"concentration" between the sunflower oil and ozonide oil so that we can compare directly. I am assuming that at least a dilution of sunflower oil was done. It should be added that information.
ANSWER 2. The percentage of ozonized oil was in the 97% interval. This information has now been added in Figure 2 legend.
COMMENT 3. Figure 3: Trypan blue not only detect death cells but all the non-viable cells. Change it on the figure and on the text.
ANSWER 3. Suggested change has been added both in Figure 3 legend and text.
COMMENT 4. Please, add values to this graph to be easily compared with figure 11
ANSWER 4. Values have been added in Figure 4 as suggested.
COMMENT 5. This merged definition needs to be clarified. It is missing one channel to show before merging one. It seems that the authors do not always present the same field to compare different staining (B and C). Missing more information on this figure.
ANSWER 5. The term ‘merged’ has been deleted both in text and Figure 6 legend. More information dealing Figure 6 have been added in the text (paragraph before Figure 6).
COMMENT 6. Timelines: I do not understand why we have so many different timelines according to the assay. As an example, Figure 1 MTS was done at 2h, 6h, 12, 24h. So when apoptosis is studied, why do not choose the same timelines, instead, a new 4h is added. Can you please justify these different timelines, different choices in all the manuscript?
ANSWER 6. Different biological events have different timelines. Microscope examination of cell morphology (Figure 2) showed that in cancer cells treated with ozonized oil, at 1 h cell viability is still maintained while cell sufferance and lack of viability is massive at 24h. Accordingly, the mechanisms causing loss of cell viability should occur in the 1-24 h time interval. Because the activation of apoptotic mechanisms requires at least 4 h, this was the timeline when we decided to evaluate this parameter. This explanation has now been added in the text (Materials and Methods, Evaluation of apoptosis, last paragraph).
COMMENT 7. line 405: How can the author conclude that there is no cell viability alteration on up to 3h incubation? How did the authors conclude there is no cytotoxicity? Did they let the cell grow and stay more than 3h with the tested oil? Was any quantification done or just observed in a light microscope?
ANSWER 7. Cell viability of keratinocytes treated with ozonized oils quantified by MTT test was 100% in ANSWER 7. control cells, 99.8% after 1h, 99.4% after 2h, 98.7% after 3h in ozonized oil treated cells. Cell viability was not evaluated at times >3 h because the oil interface blocked cell exchange with culture medium causing cell sufferance both in sham treated cells (sunflower oil) and ozonized oil treated cells independently of oil toxicity. This information has now been added in the text (Results, 3.8 Inability to kill normal cells, last paragraph).
COMMENT 8. Table 1: not very comfortable with some conclusions. For example mitotic arrest (how was it checked?) and keratinocytes-related conclusions. This table is very useful, but I suggest adding the figure to the corresponding conclusion to be easier to follow this resume table. Double-check if the study supports all your resume conclusion in this table.
ANSWER 8. The reference Figure has been added in Table 1 as suggested (newly added column on the right). The term ‘mitotic arrest’ was deleted. Table 1 was checked and revised.
COMMENT 9. Line 433, 434: "These results provide evidence that 433 HOO can induce anti-inflammatory effect without causing immunosuppression." To be clear to the reader your conclusion, you should present cell subset number/percentage in the blood analysis that was done (supplementary figure).
ANSWER 9. A paragraph reporting lymphocytes subsets (CD4+ and CD8+) evaluated by cytofluorimetric analysis further to monocytes activation (HLAdr+) in the blood analysis that was done has now been added in the text (paragraph before Figure 12).
COMMENT 10. There is no scale on the microscope figures
ANSWER 10. The scale was added in microscope Figures 2,3,5 and 10.
COMMENT 11. Amazing in vivo results! But on these follow-up patients, can you clarify which kind of follow up exams were done. Do they develop any different co morbidities? This data will be important if you have it to add as supplementary information.
ANSWER 11. Standard follow up exams mainly included hematological analyses, blood analysis of cancer markers (e.g. Ca-19), nuclear magnetic resonance, computerized tomography with and without glucose tracer, ecography. This sentence has been added in the text (paragraph before Figure 19). The analysis of this massive dataset is out of the focus of the herein presented paper (that is already very long and complex) and will be finalized in a future paper.
Reviewer 3 Report
The authors have evaluated the efficacy of HOO treatment in killing cancer cells and integrating it with the therapeutic effects of standard therapies. Despite the great interest in the topic, the paper needs some overhaul for better presentation.
- Some fundamental issues in the language introduction need to be addressed, including lack of referencing.
- There is also a significant issue with the flow of the text. Many of the sentences seem irrelevant/ unconnected, and it is often hard to understand what the author is trying to say.
- The authors have provided only 17 references; research progress in this aspect should be added in the introduction section. More discussion and analysis with previous studies can potentially support the claim made in the manuscript.
- HOO was tested on only five adult males? Why was the sample size so less? Is the sample size enough for such analysis?
- Words in italics must be used uniformly throughout the manuscript. E.g., in vitro, It must be uniformly mentioned throughout the manuscript.
- Update the literature with the latest references. Add following important references
Wani JA, Majid S, Khan A, Arafah A, Ahmad A, Jan BL, Shah NN, Kazi M, Rehman MU. Clinico-Pathological Importance of miR-146a in Lung Cancer. Diagnostics (Basel). 2021 Feb 10;11(2):274.
Radzimierska-Kaźmierczak M, Śmigielski K, Sikora M, Nowak A, Plucińska A, Kunicka-Styczyńska A, Czarnecka-Chrebelska KH. Olive Oil with Ozone-Modified Properties and Its Application. Molecules. 2021 May 21;26(11):
Author Response
Reviewer 3:
COMMENT 1. Q1. Some fundamental issues in the language introduction need to be addressed, including lack of referencing. The authors have provided only 17 references; research progress in this aspect should be added in the introduction section. More discussion and analysis with previous studies can potentially support the claim made in the manuscript
ANSWER 1 13 new references have been added 9 in Introduction, 3 in Discussion, and 1 in Table 1. Accordingly new paragraphs discussing these newly added references have been added in Introduction thus expanding the overview of the scientific background supporting the use of ozonized oil as integrated cancer therapy and prevention. The 2 suggested references have been added in Introduction.
COMMENT 2. . HOO was tested on only five adult males? Why was the sample size so less? Is the sample size enough for such analysis?
ANSWER 2. As reported in the text (lines 632-633), after the initial evaluation of 7 cancer patients for those clinical data are reported in Figures 13-17, a total of 115 cancer patients, 76 males and 39 females was evaluated. This sample size was adequate to reach the statistical significance threshold using the Chi square test (lines 646-649).
COMMENT 3. Words in italics must be used uniformly throughout the manuscript. E.g., in vitro, It must be uniformly mentioned throughout the manuscript.
ANSWER 3. Words in italics are now used uniformly as suggested.
Round 2
Reviewer 2 Report
Dear authors, good work, especially with the amazing in vivo results. I hope to see the manuscript about all the follow-up data in the future, which will be of great importance to scientists and clinicians about this possible new therapy/adjuvant therapy.